# Attachment Performance of Stick Insects (Phasmatodea) on Plant Leaves with Different Surface Characteristics

**DOI:** 10.3390/insects13100952

**Published:** 2022-10-19

**Authors:** Judith Burack, Stanislav N. Gorb, Thies H. Büscher

**Affiliations:** Department of Functional Morphology and Biomechanics, Zoological Institute, Kiel University, 24118 Kiel, Germany

**Keywords:** adhesion, leaf surface, tarsus morphology, trichomes, surface free energy, mechanoecology, ecomorphology

## Abstract

**Simple Summary:**

Herbivorous insects and plants greatly affected each other’s evolution due to their close interactions. This resulted in the development of a variety of adaptations on both sides. Through the need for protection against herbivorous insects, surfaces with lower attachment ability evolved in many plants. As a response, the attachment systems of insects have developed numerous specializations. Stick insects (Phasmatodea) have an attachment system, consisting of paired claws, arolium (attachment pad between the claws) and euplantulae (paired attachment pads on the tarsomeres), which is well adapted to different natural surfaces. We used measurements of pull-off and traction force in two species (*Medauroidea extradentata* and *Sungaya inexpectata*) representing the most common microstructures used for attachment within stick insects (nubby and smooth) to quantify the attachment ability of Phasmatodea on natural surfaces. Plant leaves with different surface properties (smooth, trichome-covered, hydrophilic and covered with crystalline waxes) were selected as substrates. Wax-crystal-covered fine-roughness substrates revealed the lowest, whereas strongly structured substrates showed the highest attachment performance among the stick insects studied. Removing the claws of the insects resulted in lower attachment ability on structured substrates. Furthermore, claw removal revealed that the attachment performance of the pads is less reduced by contaminating wax crystals in the species with nubby attachment structures. Long-lasting effects of the leaves on the attachment ability were briefly investigated, but not confirmed.

**Abstract:**

Herbivorous insects and plants exemplify a longstanding antagonistic coevolution, resulting in the development of a variety of adaptations on both sides. Some plant surfaces evolved features that negatively influence the performance of the attachment systems of insects, which adapted accordingly as a response. Stick insects (Phasmatodea) have a well-adapted attachment system with paired claws, pretarsal arolium and tarsal euplantulae. We measured the attachment ability of *Medauroidea extradentata* with smooth surface on the euplantulae and *Sungaya inexpectata* with nubby microstructures of the euplantulae on different plant substrates, and their pull-off and traction forces were determined. These species represent the two most common euplantulae microstructures, which are also the main difference between their respective attachment systems. The measurements were performed on selected plant leaves with different properties (smooth, trichome-covered, hydrophilic and covered with crystalline waxes) representing different types among the high diversity of plant surfaces. Wax-crystal-covered substrates with fine roughness revealed the lowest, whereas strongly structured substrates showed the highest attachment ability of the Phasmatodea species studied. Removal of the claws caused lower attachment due to loss of mechanical interlocking. Interestingly, the two species showed significant differences without claws on wax-crystal-covered leaves, where the individuals with nubby euplantulae revealed stronger attachment. Long-lasting effects of the leaves on the attachment ability were briefly investigated, but not confirmed.

## 1. Introduction

Insects and plants have interacted for a long period of time, leading to coevolution that resulted in diverse interactions between them [1,2,3,4]. On the plant side, the attraction of pollinators or vectors for seed dispersal plays a role, but also mechanisms for protection against herbivorous insects have evolved [3,4,5]. The plants’ defensive strategies against damage due to herbivory or oviposition vary from chemical defense to structural surface modifications associated with mechanical protection [6,7]. While some insects developed specialized attachment systems, plants responded accordingly with attachment-diminishing strategies [1,7]. Evolutionary novel acquisitions do not necessarily fulfill only one single purpose, but can be a result of different selective pressures. The following characteristics have been found to serve as anti-attachment surface modifications. Some plant surfaces make use of modified cell shape, cell orientation or wet coverage as it can be found in the pitcher rim of some *Nepenthes* species (Nepenthaceae) [4,8]. Trichomes of some plants are also known to reduce the attachment ability of insects to the plant surface by decreasing the actual contact area for the feet. Glandular trichomes are capable of either chemically poisoning the insect or mechanically impeding its movement ability by various secretions [9,10,11]. However, there are also observations of trichomes increasing the attachment ability of insects by providing an additional ‘foothold’, e.g., [4,12,13,14]. Another plant surface characteristic that is also reported to affect insect attachment is cuticular folds [15]. Similar to trichomes, they can either reduce the contact area or increase the potential interlocking sites for claws depending on insect species [13]. A very effective way to decrease attachment ability of insects on plant surfaces are epicuticular wax crystals [4,15,16]. These specialized surface coatings are found, for example, on the leaves of *Eucalyptus* species (Myrtaceae) and presumably cause loss of adhesion through increased roughness and impairment of the attachment system by contamination, wax-dissolving or fluid-adsorption [5,17]. As shown for some insects, wax crystals can contaminate the attachment system at first contact, but they can be subsequently removed during the next footsteps [4,17].

Previous studies on the attachment of insect species on different plant surfaces showed that the surface of plants affects the attachment of insects. For example, the bug *Nezara viridula* (Linnaeus, 1758) (Heteroptera: Pentatomidae) showed a significantly lower attachment ability on leaves covered with cuticular waxes compared to leaves with a trichome coverage [6]. Salerno et al. [18] showed a stronger attachment of two ladybird species onto hydrophilic surfaces compared to hydrophobic ones [18]. Voigt et al. [13] investigated the attachment ability of *Dicyphus errans* (Wolff, 1804) (Heteroptera: Miridae) on the surfaces of six different plants. This bug showed reduced attachment to plant surfaces with wax crystals and strong attachment on plant surfaces with either nonglandular or glandular trichomes. Consequently, most literature sources conclude that the plant-surface type has a significant influence on the insect attachment ability [13,19,20].

The focal insect group in this study is Phasmatodea, which are also known as stick and leaf insects. It is a lineage of insects with more than 3400 known species and worldwide distribution [21,22]. Stick and leaf insects are herbivorous and many camouflage themselves as parts of plants, such as bark, twigs, leaves or moss [23]. Phasmids inhabit most habitats in various tropical and subtropical ecosystems and are distributed from the ground to the canopy within forests [24]. Owing to close coevolution with plants and due to the different conditions of substrates in the respective environments, different adaptations to attachment evolved within Phasmatodea. The ground pattern of phasmid attachment structures always include two types of attachment pads, the pretarsal arolium and the tarsal euplantulae, together with paired claws. [25]. The two pads work in complementary directions: the arolium provides adhesion to the substrate and the euplantulae generate friction due to shear forces when pressed onto the surface. The combination of these pads and the adjustability of the amount of involved euplantulae results in a highly adaptive attachment system [25,26,27,28,29,30]. While the arolium of Phasmatodea is usually smooth, the euplantulae show a high diversity of microscopic surface structures [25], most likely to adapt best to the corresponding substrate conditions prevailing in different habitats. The most common microstructures are smooth and nubby, whereas other euplantula microstructures are characterized by different pattering and aspect ratios resulting in several potential functions, such as frictional anisotropy, randomization of pattern directionality and coping with water or particular contaminations [25,28,29]. The adhesion and traction of the nubby and smooth euplantulae have already been experimentally tested on artificial surfaces with different roughness. These studies showed that smooth euplantular microstructures provide better attachment on smooth surfaces and nubby euplantular microstructures perform comparatively better on microrough surfaces [26,27,31]. It has been shown that the claws are important for interlocking with rough surfaces and work complementarily with the attachment pads [26,32]. The experimental studies on stick insect attachment so far comprise exclusively artificial substrates to elucidate the basic functionality of the attachment system of these insects. Plants, however, offer various influences on the attachment performance on the one hand, and are the most important substrate for these herbivores.

The aim of this study is to investigate the attachment ability of Phasmatodea on plant surfaces with different characteristics. Specifically, we try to answer, whether there is a difference in attachment ability on natural substrates between two species with different attachment microstructures: *Medauroidea extradentata* (Brunner von Wattenwyl, 1907) (Phasmatidae) with smooth and *Sungaya inexpectata* (Zompro, 1996) (Heteropterygidae) with nubby euplantula microstructure. We used attachment force measurements to compare (I) if the attachment ability is different on the leaves of the four selected plant species, (II) if there are differences between the performances of the two species and (III) whether the performance was influenced by removing the claws. Furthermore, we explored a potential long-lasting contamination of the attachment system by plant surfaces.

## 2. Materials and Methods

### 2.1. Species

We used the same two species as used in Büscher and Gorb [26], because of their difference in euplantular microstructure [26]. *Medauroidea extradentata* has a smooth arolium and smooth euplantulae and *Sungaya inexpectata* has nubby euplantulae and a smaller smooth arolium (Figure 1) [25,26]. *Sungaya inexpectata* has an accessory euplantula on the fifth tarsomere (Figure 1A), which is found in some phasmid species [26]. The animals were taken from the laboratory cultures of the Department of Functional Morphology and Biomechanics (Kiel University, Kiel, Germany). The age of the individuals was not further considered, as they were selected by adequate size oriented to the used leaves (mean body mass: *M. extradentata* 144 ± 69 mg; *S. inexpectata* 332 ± 174 mg). During the time of the experiment, the animals were kept in adequate boxes: 10–20 animals and fed with hazelnut and blackberry leaves ad libitum. Each individual was checked for intactness of legs and tarsus before the measurements and was replaced in case of damage.

### 2.2. Experimental Substrates 

The leaves of *Epipremnum aureum* (Linden et André) (Alismatales: Araceae), *Tibouchina urvilleana* (Cogniaux) (Myrtales: Melastomataceae), *Hoffmannia ghiesbreghtii* (Lemaire) (Gentianales: Rubiaceae) and *Eucalyptus globulus* (Labillardière) (Myrtales: Myrtaceae) were used as substrates for the attachment of the tested animals. They were chosen because of their surface characteristics and for their suitable leaf size. *Epipremnum aureum* (Figure 2A,B) has leaves with a smooth glossy to sometimes dull lamina, the upper side of the blade [33]. The leaves of *T. urvilleana* (Figure 2C,D) are abundantly covered with nonglandular trichomes [34]. *Hoffmannia ghiesbreghtii* (Figure 2E,F) has leaves that are glabrous on the adaxial side [35] and also very hydrophilic (Figure 2F). *Eucalyptus globulus* leaves (Figure 2G,H) have a smooth surface covered with wax in different amounts depending especially on the age of the leaves and different components [36,37]. A glass plate represented the control substrate. The leaves were collected on the day of use from the Botanical Garden Kiel (Christian-Albrechts-University Kiel, Kiel, Germany). It was always assured that the leaves were intact and in a fresh state.

### 2.3. Force Measurement 

The force measurement setup (Figure 3) followed the one used by Büscher and Gorb [26] and Büscher et al. [38] for similar measurements. A BIOPAC Model MP100 and a BIOPAC TCI-102 system (BIOPAC Systems, Inc., Goleta, CA, USA) connected to a force transducer (25 g capacity; FORT25, World Precision Instruments Inc., Sarasota, FL, USA) were used for the force measurements. The plant leaves were fixed with double-sided adhesive tape on a glass board adaxial side up and held in a fresh state with a wetted paper towel secured with parafilm during the experiment. Throughout the whole experiment, the temperature ranged between 21 and 24 °C and the ambient humidity between 17 and 52%.

The experimental procedure was identical for all species and substrates. First, an insect was randomly chosen, weighed and anesthetized with CO_2_. While sedated, a human hair with a loop on one side was attached using a drop of melt wax onto the metanotum. The loop of the hair was then secured onto the force transducer. After full recovery of the stick insect, the force measurement was started. For pull-off force measurements, the force transducer was aligned perpendicular to the substrate and the substrate was constantly lowered with an approximate speed of 0.5–1.0 cm/s through a laboratory scissor jack until the insect completely lost contact to the surface. The traction force was measured by aligning the force transducer horizontal to the substrate. The phasmid was held in position facing the stem of the leaf, while the surface was constantly pulled away from the transducer so that the animal was pulled along its body axis (approx. 200–300 µm/s) until the individual completely stretched the legs and showed loss of grip. The force–time curves were recorded with AcqKnowledge 3.7.0 software (BIOPAC Systems Inc., Goleta, CA, USA). The value of the highest peak of the curves represented the maximum pull-off or traction force (F_Ind,Sub_). For both measurements, the measuring was repeated three times per individual and the mean value was calculated to reduce intraindividual variability. Every phasmid was given sufficient resting time in between the measurements. To generate a value independent of the body mass (m_Ind_) of the individuals, the safety factor for pull-off and traction force was calculated with the following equation, by normalizing the force by the insect’s body mass (1):Safety factor_Ind,Sub_ = F_Ind,Sub_ × (m_Ind_ × g)^−1^(1)

These measurements with no further specifications were done with 15 individuals (*n* = 45, *N* = 15) of each species. 

#### 2.3.1. Claw Manipulations

The above measurements were also repeated with 10 individuals of both species with amputated claws for the minimization of potential interlocking between the substrate and the claws. The claws were cut off just above the claw base with fine scissors while the animal was sedated with CO_2_ and attention was paid to not injure the arolium or other parts of the attachment system.

#### 2.3.2. Contamination Effects

To find possible long-lasting effects of the plant leaves on the attachment ability of each species, another set of force measurements was performed. In consideration of potential effects due to the wax crystals of *Eu. globulus,* the lasting measurement on those leaves was done with 15 individuals of *M. extradentata* and 15 individuals of *S. inexpectata*. The other substrates (glass, *Ep. aureum*, *T. urvilleana*, *H. ghiesbreghtii*) were tested with 5 individuals of each species. All individuals were prepared as previously described, but were tested with a different order of substrates: first, the force was measured once on glass as a reference, then once on the respective plant substrate (*Ep. aureum*, *T. urvilleana*, *H. ghiesbreghtii* or *Eu. globulus*) and subsequently measured twice on glass. These two last measurements were pooled in the further data processing. This was conducted only once with each individual for pull-off and traction force on each substrate. The glass was wiped with a paper towel after each individual. Based on the obtained values, the safety factors (e.g., SF_glass1_, SF*_Ep.aureum_*, SF_glass2_, SF_glass3_) were similarly calculated in the other experiments.

An additional control measurement with only glass as substrate was performed for both species.

### 2.4. Claw Metrics

The size of the claws was measured by dissecting them in five individuals of *M. extradentata* and *S. inexpectata*. The claws were air dried, mounted to aluminum stubs and coated with 10 nm thickness gold–palladium. Claws were studied by using a scanning electron microscope (SEM; Hitachi TM3000, Hitachi High-technologies Corp., Tokyo, Japan) at 15 kV acceleration voltage. The inner claw curvature (r_in_) and the claw tip sharpness (d_tip_) were measured with ImageJ [39], following Büscher and Gorb [26]. 

### 2.5. Surface Characterization

To characterize the surface of the substrates, the roughness and the height of leaf veins were measured with the optical surface scanner macroscope (VR-3000 Series, Keyence, Osaka, Japan). For this purpose, seven to ten areas on the midvein (*Ep. aureum*, *Eu. globulus*: 10, *T. urvilleana*: 7, *H. ghiesbreghtii*: 8) and ten areas on secondary veins on the adaxial side of one leaf per species were selected (Appendix A). We measured arithmetic mean roughness (R_a_), maximum profile peak height (R_p_), maximum profile valley depth (R_v_) and maximum roughness (R_z_). For *Ep. aureum* and *Eu. globulus,* a high magnification (40×) was used, whereas for *T. urvilleana* and *H. ghiesbreghtii* a lower magnification (12×) was used due to the more pronounced surface geometry. The adaxial side of the leaves was examined for further characterization of the surface structures by using the SEM Hitachi S-4800 (Hitachi High-Technologies Corp., Tokyo, Japan) equipped with a Gatan ALTO 2500 cryopreparation system (Gatan Inc., Cambridge, UK). See Gorb and Gorb [40] for details on the preparation method. Whole mounts of the leaves were sputter-coated with 10 nm gold–palladium while frozen and examined in the cryostage of the microscope at 3 kV acceleration voltage and a temperature of −120 °C.

### 2.6. Statistical Analysis

For the statistical analysis, SigmaPlot 12.0 (Systat Software Inc., San José, CA, USA) was used. All results were tested for normal distribution (Shapiro–Wilk test) and, if passed, tested with an Equal Variance Test (Levene). The means of the measurements of the two species on the plant leaves were compared using Kruskal–Wallis one-way analysis of variance (ANOVA) on ranks and Tukey’s post hoc test with a significance level of 0.05. The results obtained on the claw-amputated individuals were analyzed using Kruskal–Wallis ANOVA followed by all pairwise multiple comparison procedures (Dunn’s method) within the species or Tukey’s post hoc test between the two species. The data of the contamination measurement were tested with one-way ANOVA and all pairwise multiple comparison procedures (Holm–Šídák method) or Kruskal–Wallis ANOVA and Dunn’s method accordingly to the distribution.

## 3. Results

### 3.1. Force Measurements

The attachment performance of *M. extradentata* and *S. inexpectata* (Figure 4) revealed higher traction force than pull-off force for both species on all substrates. 

The pull-off measurement (Figure 4A) showed the highest mean for both phasmid species on *H. ghiesbreghtii* with 11.814 ± 4.840 (mean ± s.d.) for *M. extradentata* and 13.287 ± 13.057 for *S. inexpectata*. The second highest and statistically not different pull-off forces were measured on *T. urvilleana* for both species. For the pull-off measurement on *Ep. aureum* and *Eu. globulus,* mean values of 3.064 ± 1.208 and 1.357 ± 0.220 occurred for *M. extradentata* and mean values of 4.212 ± 1.658 and 1.523 ± 0.827 for *S. inexpectata*. The pull-off force measurement on glass with *M. extradentata* revealed a mean statistically different from all other substrates besides the measurement on *Ep. aureum* with a value of 5.014 ± 1.774. The control glass with *S. inexpectata* revealed a mean of 7.945 ± 4.716, only statistically different to the measurement of this insect species on *Eu. globulus*. The pull-off measurements of *M. extradentata* were not statistically different between *T. urvilleana* and *H. ghiesbreghtii*, and between *Ep. aureum* and glass. All other combinations differ significantly from each other. The pull-off force of *S. inexpectata* between the measurements on *Eu. globulus* and each substrate and between *Ep. aureum* and each substrate, except glass, were statistically different. All other combinations showed no significant differences for *S. inexpectata*. (H = 287.832, d.f. = 9, *N* = 15, *p* ≤ 0.001; Tukey’s test, *p* < 0.05).

In the traction force measurements (Figure 4B), the highest traction force of *M. extradentata* was reached on the leaves of *H. ghiesbreghtii* with a mean of 29.300 ± 10.951 and for *S. inexpectata* on the same substrate with a mean of 34.514 ± 19.799. On *E. globulus,* the lowest traction force was obtained with a mean of 1.936 ± 2.475 and 5.736 ± 5.451 of *M. extradentata* and *S. inexpectata*. The leaves of *Ep. aureum* and *T. urvilleana* and the glass control revealed a mean of 11.731 ± 4.661, 22.410 ± 5.207 and 19.952 ± 7.520 for the traction force of *M. extradentata*. For *M. extradentata,* there were no significant differences among glass, *T. urvilleana* and *H. ghiesbreghtii*. This is similar with *S. inexpectata*, with means of 10.067 ± 5.039 on *Ep. aureum*, 24.631 ± 12.885 on *T. urvilleana* and 26.828 ± 11.164 on glass. Solely the measurements on *Ep. aureum* were not significantly different compared to the traction force on *Eu. globulus*. (H = 269.977, d.f. = 9, *N* = 15, *p* ≤ 0.001; Tukey’s test, *p* < 0.05).

The comparison of the pull-off and traction force between the two insect species (Figure 4) revealed that the values of *S. inexpectata* are slightly higher than those of *M. extradentata*, except by *T. urvilleana* and *H. ghiesbreghtii,* as substrates for the pull-off force measurement and *Ep. aureum* as the substrate for the traction force measurement. The statistical analysis showed no difference between *M. extradentata* and *S. inexpectata* on any tested substrate for both types of force measurements (pull-off: H = 287.832, d.f. = 9, *N* = 45, *p* ≤ 0.001; Tukey’s test, *p* < 0.05; traction: H = 269.977, d.f. = 9, *N* = 15, *p* ≤ 0.001; Tukey’s test, *p* < 0.05).

#### 3.1.1. Force Measurement after Claw Manipulations

The individuals without claws of *M. extradentata* showed overall lower attachment force values for the pull-off force measurements, if compared with individuals of the same species with intact claws (Figure 5A). The highest mean pull-off force, however, was similar to the measurements with claws on the leaves of *H. ghiesbreghtii* (3.584 ± 1.791) followed by the mean measured on *T. urvilleana* (2.999 ± 0.903), on glass (2.356 ± 0.821), on *Ep. aureum* (1.530 ± 0.432) and the lowest mean pull-off force on *Eu. globulus* (0.910 ± 0.097). The statistical analysis revealed a significant difference between the individuals with and without claws on the glass control, *T. urvilleana* and *H. ghiesbreghtii* for the pull-off of *M. extradentata* (Kruskal–Wallis one-way ANOVA on ranks, H = 283.535, d.f. = 9, N_without_claws_ = 10, N_with_claws_ = 15, *p* ≤ 0.001; Dunn’s method, *p* < 0.05). 

In the case of the pull-off force measurement of *S. inexpectata,* the individuals with claws also showed higher values except on the leaves of *Eu. globulus* (Figure 5B) and the same ranking order as *M. extradentata*. The only statistical differences were found between the individuals with and without claws on glass and *T. urvilleana* (H = 179.328, d.f. = 9, N_without_claws_ = 10, N_with_claws_ = 15, *p* ≤ 0.001; Dunn’s method, *p* < 0.05). 

The traction forces of *M. extradentata* were also higher for individuals with intact claws (Figure 5C). However, different from the measurement with intact claws, the highest mean for individuals without claws occurred on glass (11.800 ± 8.524) and not on *H. ghiesbreghtii* (7.034 ± 5.374). The lowest traction was found on *Eu. globulus* with a mean of 0.359 ± 0.823. The values for *M. extradentata* with and without claws were statistically significantly different on glass, *T. urvilleana* and *H. ghiesbreghtii* (H = 276.410, d.f. = 9, N_without_claws_ = 10, N_with_claws_ = 15, *p* ≤ 0.001; Dunn’s method, *p* < 0.05).

*Sungaya inexpectata* without claws had mostly lower traction force, if compared with individuals of the species with intact claws (Figure 5D). However, the mean on *Ep. aureum* (12.352 ± 7.596) and *Eu. globulus* (6.874 ± 3.266) was higher for *S. inexpectata* individuals without claws. There was a significant difference between the traction force of *S. inexpectata* with and without claws on *H. ghiesbreghtii*, *T. urvilleana* and glass, but not between *Ep. aureum* and *Eu. globulus* (H =171.182, d.f. = 9, N_without_claws_ = 10, N_with_claws_ = 15, *p* ≤ 0.001; Dunn’s method, *p* < 0.05)

The comparison of individuals without claws between the two species revealed higher values of pull-off and traction force for *S. inexpectata* (Figure 5E,F). Nevertheless, the only significant difference between *M. extradentata* and *S. inexpectata* was found on *Eu. globulus* for both types of force measurements (traction and pull-off). Between all other substrates, no significant differences were found (H_pull-off_ = 126.899, H_traction_ = 100.812, d.f. = 9, *N* = 10, *p* ≤ 0.001; Tukey’s test, *p* < 0.05).

#### 3.1.2. Contamination Effects

The measurements testing the lasting of effects toward the attachment system of *M. extradentata* (Figure 6) showed no statistical difference between the values on glass before and after the substrate for any of the plant species studied, except for the pull-off measurement with *Ep. aureum.* The pull-off force on *Ep. aureum* and glass 2 were statistically not different, the first measurement on glass was significantly different from both of them (one-way ANOVA, F = 8.453, d.f. = 2, N_glass1,Ep.aureum_ = 5, N_glass2_ = 10, *p* = 0.003; Holm–Šídák method, *p* < 0.05). For the traction measurement with *M. extradentata* on *Ep. aureum* and the pull-off measurements on *T. urvilleana* and on *H. ghiesbreghtii,* all the means were not significantly different from each other (one-way ANOVA, F < 2.76, d.f. = 2, N_glass1,plants_ = 5, N_glass2_ = 10, *p* < 0.37). The traction force measured on the plant leaves of *T. urvilleana* and *H. ghiesbreghtii* were significantly different from the traction force on glass, but glass 1 and glass 2 did not differ significantly from each other (*T. urvilleana*: one-way ANOVA, F = 7.722, d.f. = 2, N_glass1,T.urvillena_ = 5, N_glass2_ = 10, *p* = 0.004; Holm–Šídák method, *p* < 0.05; *H. ghiesbreghtii*: Kruskal–Wallis one-way ANOVA on ranks, H = 1.054, d.f. = 2, N_glass1,H.ghiesbreghtii_ = 5, N_glass2_ = 10, *p* = 0.590). Likewise, the pull-off and traction force at the measurement with *Eu. globulus* on glass 1 and glass 2 were not significantly different from each other; the values on *Eu. globulus* were significantly different to the values on glass (Kruskal–Wallis one-way ANOVA on ranks, H < 21.68, d.f. = 2, N_glass1,Eu.globulus_ = 15, N_glass2_ = 30, *p* ≤ 0.001; Dunn’s method, *p* < 0.05).

The measurement with *S. inexpectata* testing for contamination effects to the attachment system (Figure 7) revealed for neither of the substrates a statistical difference between the values on glass before and after the leaf. In the pull-off measurement with *S. inexpectata* on *Ep. aureum*, on *T. urvilleana* and each measurement on *H. ghiesbreghtii,* the mean values were not statistically different from each other (Kruskal–Wallis one-way ANOVA on ranks, H < 4.82, d.f. = 2, N_glass1,plants_ = 5, N_glass2_ = 10, *p* ≤ 0.91). The traction measurement with *Ep. aureum* in-between glass, the values on glass 1 and glass 2 and the values on glass 1 and *Ep. aureum* were statistically not different from each other, while those on *Ep. aureum* and glass 2 were significantly different (one-way ANOVA, F = 4.013, d.f. = 2, N_glass1,Ep.aureum_ = 5, N_glass2_ = 10, *p* = 0.037; Holm–Šídák method, *p* < 0.05). The traction force on glass 1 and glass 2 in the measurements on *Eu. globulus* and *T. urvilleana* were not significantly different from each other. The value on the plant leaves were statistically different from the measurements on glass (*Eu. globulus*: Kruskal–Wallis one-way ANOVA on ranks, H < 25.81, d.f. = 2, N_glass1,Eu.globulus_ = 15, N_glass2_ = 30, *p* ≤ 0.001; Dunn’s method, *p* < 0.05; *T. urvilleana*: one-way ANOVA, F = 5.927, d.f. = 2, N_glass1,T.urvillena_ = 5, N_glass2_ = 10, *p* = 0.011; Holm–Šídák method, *p* < 0.05).

The additional control measurement with only glass as substrate revealed no significant differences between the attachment ability on the three glasses for both species (Appendix A).

### 3.2. Leaf Surface Characteristics 

The surfaces of the adaxial sides of the leaves used as substrates were visualized using cryoscanning electron microscopy (Figure 8). The surface of *Ep. aureum* was the smoothest and revealed almost no surface protrusions. However, it contains a rather thick but smooth wax layer (Figure 8A,B). *Tibouchina urvilleana* leaves are more structured and the surface is densely covered with elongated trichomes with a length of 432.6 ± 151.3 µm (mean ± SD, *n* = 10) with pointed tips (Figure 8C,D). The adaxial leaf surface of *H. ghiesbreghtii* has an undulating profile and convex hemispherically shaped cells. Occasionally trichomes are found at the leaf edge (Figure 8E,F). Additionally, this surface was partially fouled with microorganisms. The adaxial surface of the leaves of *Eu. globulus* is covered with wax crystals. These wax protrusions are tubular and elongated, with a length of 3.1 ± 1.4 µm (Figure 8E,F).

The surface texture of the substrates was characterized by measurements of the roughness and the vein height on different areas on the adaxial side of the leaves (Appendix A). Figure 9A shows the arithmetical mean roughness R_a_ and Figure 9B shows the heights of the middle and secondary vein of each leave combined. The leaf of *H. ghiesbreghtii* shows the highest arithmetical mean roughness (514.5 ± 146.7 µm) and mean vein height (1470.5 ± 997.4 µm). With less than a tenth of the mean R_a_ of *H. ghiesbreghtii, Eu. globulus* has the lowest arithmetical mean roughness with 44.1 ± 22.1 µm and with 109.0 ± 80.4 µm, which is also the lowest mean value of vein heights. The leaves of *T. urvilleana* and *Ep. aureum* show intermediate roughness compared to the previous two species, whereas *T. urvilleana* has a higher mean R_a_ (106.2 ± 49.3 µm) and mean vein height (408.6 ± 184.2 µm) than *Ep. aureum* with a mean vein height of 309.5 ± 196.6 µm and mean R_a_ of 90.3 ± 53.4 µm.

## 4. Discussion

### 4.1. Attachment Performance on the Different Plant Leaves

We investigated the influence of different plant leaf characteristics on the attachment performance of stick insects. In general, the performance varied across the leaves used as substrates. The highest pull-off and traction forces were reached on the microstructured, hydrophilic leaves of *H. ghiesbreghtii* followed by the values on the trichome-covered leaves of *T. urvilleana*. Both substrates are relatively rough and are either covered with trichomes (*T. urvilleana*, Figure 8C,D) or present an undulating surface profile due to the leaf venation (*H. ghiesbreghtii*, Figure 8E,F). The elevations of the leaves of *H. ghiesbreghtii* seemed to generate a better holding surface for phasmids. Phasmids already showed an increased pull-off force on curved artificial substrates, if compared to flat surfaces [38]. Similar effects have been observed in other insects [41,42,43,44,45] and frogs [46,47]. In contrast to all other leaves on which the phasmids engaged the arolium only during the pull-off measurement, and similar to the individuals measured on flat artificial substrates in Büscher and Gorb [26], some individuals brought their euplantulae into contact during the pull-off measurement on *H. ghiesbreghtii*. The waviness of the surface additionally has an influence on the peeling angle of the tarsus during detachment and consequently on the resulting attachment force [38,48]. Furthermore, the leaves of *H. ghiesbreghtii* seem to be less stiff and might enable penetration of stiff claws. The claws of *S. inexpectata* were observed to cause small scratches during the traction measurement. The hydrophilic properties of the leaves enhanced adhesion of Phasmatodea species studied [49]. Phasmatodea have adhesive secretions [50,51,52,53] that could be positively affected by the hydrophilicity of the substrate [49]. 

Trichomes on plant surfaces have already been shown to generate an additional “foothold”, thereby increasing the insect attachment, e.g., [12,13,16,17,54]. The trichomes of *T. urvilleana* provide good support for mechanical interlocking of the claws because they are long enough to represent such a “foothold” for the tested individuals of both species. The attachment ability on the relatively smooth leaves of *Ep. aureum* (Figure 8A,B) of both species was lower than on *H. ghiesbreghtii* and *T. urvilleana.* The pull-off and traction force on *Ep. aureum* were generally lower than on glass, but the difference was found significant only for the traction measurement with individuals with intact claws. The slightly lower attachment performance can be due to the different surface free energy and presumably lower stiffness of *Ep. aureum* leaves compared to glass. Studies showed higher attachment force on smooth substrates generally, explained through a high real contact area compared to rough surfaces, e.g., [13,17,54,55,56,57,58,59], since the claws cannot interlock on smooth substrates [26,32,41]. In the case of Phasmatodea, the force produced by successful mechanical interlocking of the claws surpasses the attachment by the attachment pads, at least in the case of pronounced trichomes of *T. urvilleana*. 

The lowest traction and pull-off force, and therefore the lowest attachment of phasmids tested herein were measured on the leaves of *Eu. globulus*. Those leaves are covered with wax crystals, but smooth underneath the wax (Figure 8G,H). They showed by far the most significant differences to the other substrates. The leaves were also the stiffest ones in compression (pers. observation) and did not show any sites for claw interlocking. The insect attachment on surfaces with epicuticular waxes was analyzed in previous studies with most studies showing a low attachment performance, e.g., [6,16,17,48,49,60,61]. This can be due to the increased roughness, impairment of the attachment system with contamination, wax-dissolving or fluid absorption [5,17,62]. During the measurements of this study, the Phasmatodea showed the most effort to get off the leaves of *Eu. globulus*, if compared to the other substrates. This may also be a behavioral response to the reduced attachment ability on the leaves. We found no indication for an effect of ambient humidity during the measurements, although such an effect is reported for other insects [50,63] did not find an effect of ambient humidity on stick insect attachment as well. However, very high ambient humidity could have an effect, but was not reached during our experiments. The effect of ambient humidity could be experimentally tested in a systematic way as it has been done for presence of water films and wettability of the substrate for phasmids [49,62].

### 4.2. Impact of the Claws 

Especially on leaves with strongly expressed surface structure, claws of insects are able to interlock and provide strong attachment. The interlocking of the claws is dependent on their structure, e.g., tip diameter, density and curvature [26,32,54,64]. In this study, the statistical difference between the attachment ability of individuals with intact claws and individuals without claws for both species was measured on glass and *T. urvilleana* for traction and pull-off force. On *H. ghiesbreghtii,* every measurement except pull-off with *S. inexpectata* was significantly different as well. Consequently, the effect of the claws has a greater impact on structured surfaces such as trichome-covered (*T. urvilleana*, Figure 8C,D) or leaves with strongly expressed microstructure (*H. ghiesbreghtii*, Figure 8E,F). 

Trichomes occur on the leaves of various plant lineages and serve different functions [65]. For the attachment ability of insects, however, the trichomes can potentially generate a “foothold” that the claws can grab onto, e.g., [12,13,16,17,54]. Nevertheless, the role of trichomes for insect attachment is also a question of the dimension of the plant feature in relation to the insect attachment system or even the insect itself. In general, trichomes are reported to hinder locomotion and negatively affect the oviposition, acceptance of the plant as food or attachment to the surface [66,67,68,69,70], but these effects can be different for differently sized insects. Insects which are small enough can simply step between trichomes [14], while a suitable size difference between trichome and tarsal claws can also lead to specialized clamping devices on the claws that use the trichomes for interlocking [71]. Clamping fibrillar surface features can strongly increase the attachment, as shown for parasitic insects, e.g., [72]. Stick insects are some of the largest insects and are several times larger than the insects reported to be repelled by plant puberance and also the tarsi of stick insects are usually much larger than the trichomes. Some aschiphasmatine stick insects possess pectinate claws as well, e.g., [30,73], but the two species investigated herein have simple pointed claws (Figure 1). Removing these revealed the noted influences, i.e., on trichome-bearing as well as corrugated surfaces and on glass. In our two focal stick insect species, the inner radius of claw curvature (Appendix A) is larger in both species than the thickness of the trichomes (Figure 8). Accordingly, the claws can grasp around single trichomes. The mechanical interlocking of claws at higher roughness or structured surfaces was already shown in other studies, e.g., [26,60,74]. For example, the removal of claws of the beetle *Chrysolina polita* (Chrysomelidae, Coleoptera) resulted in lower pull-off forces on cloth but not on glass [54]. However, our measurements presented in this study show that the attachment performance of individuals without claws was also reduced on glass for both species. This could be due to the mechanical support of the claws to the arolium. The claws provide stability and proper placement of the arolium; therefore, in their absence, the attachment ability of the arolium can be reduced [75]. 

### 4.3. Comparison of Attachment Performance between the Two Species 

The main difference between attachment systems of the two species is that *M. extradentata* has smooth euplantulae and *S. inexpectata* has nubby microstructures on the euplantulae [25]. It has been shown in previous studies that a smooth euplantula provides stronger adhesion to smooth surfaces compared to a euplantula with a nubby microstructure [26,31]. Büscher and Gorb [26] also showed that the traction force was higher for *S. inexpectata* at different defined roughnesses (0.3, 1, 12, 425 µm) and *M. extradentata* performed better on smooth and coarse substrates in the pull-off experiment [26]. In contrast to the previous measurements on standardized surfaces, the performance of the two species on natural leaves did not differ statistically on most substrates. Actual plant surfaces have a wide spectrum of different structures and properties. For example, the hydrophilicity of *H. ghiesbreghtii* leaves can affect the attachment ability of both phasmids independent of their adaptations toward different roughness. Solely the wax-crystal-covered leaves of *Eu. globulus* revealed a significant difference between individuals of *M. extradentata* and *S. inexpectata* without claws (Figure 5E,F). The attachment force of *S. inexpectata* with nubby euplantulae was significantly higher than that for the species with smooth euplantulae. Consequently, the nubby attachment pads seem to be less affected by the contamination of the wax crystals. Nubby or hairy attachment structures have already been assumed to be less influenced by contamination than smooth attachment structures [74]. When the function of the claws is excluded, the attachment pads are the main organ involved in attachment of Phasmatodea [25,26]. Species with nubby euplantula microstructures within Phasmatodea are more frequently found in species that are somewhat ground-associated, which are either known to dwell on the ground, feed on dropped leaves or bury their eggs into the soil [25,28,30]. This preference could be accompanied by potential contaminations from the soil, at least in species using their tarsi to dig holes for egg deposition (e.g., some Heteropterygidae [76]). Further experiments testing the influence of contaminations on both types of euplantula microstructures would be helpful to investigate the role of attachment microstructure for contaminations.

### 4.4. Lasting of Contaminating Effects 

The coverage of plant leaves, such as waxes [25] and excretions of glandular trichomes, e.g., [65,77,78,79] can potentially reduce the attachment ability of insects instantaneously, but often also cause persistent contaminations that last longer and remain on the feet of the insects [80]. None of the substrates used in this study seem to have a long-lasting effect on the attachment of Phasmatodea. For *Eu. globulus*, a possible persistent decrease in attachment due to contaminating effect of wax crystals on the surface of the leaves [4] could not be observed. However, this observation does not exclude the presence of some contamination. The effect of wax-crystal contamination could already be overcome before or during the first contact during the force measurement. The wax crystals of *Eu. globulus* are large in comparison to the attachment system of the phasmid studied and therefore might enable a quick removal of them. It is quite likely that the initial attachment forces might have been reduced, but with the repeating contact during walking, the attachment is regenerated due to shedding off the wax crystals. The only noteworthy difference due to possible lasting effects of the substrate was observed in the pull-off force measurement with *M. extradentata* on leaves of *Ep. aureum* in between two measurements on the glass as a control. The reason for the significant difference between the control measurements on glass with *Ep. aureum* in between could be a result of other plant leaf secretions. 

## 5. Conclusions

The attachment of Phasmatodea is influenced by different features on the surface of leaves. 

(I)The highest attachment ability is observed on the microstructured, hydrophilic leaves of *H. ghiesbreghtii* and the trichome-covered leaves of *T. urvilleana*. Strong surface corrugations and high substrate waviness are beneficial for the function of both claws and their combination with attachment pads. Epicuticular wax crystals on the surface of *Eu. globulus* leaves caused the lowest attachment ability for the Phasmatodea species studied.(II)The claws of the insects did have the strongest impact on the attachment on leaves with trichomes or strong surface corrugations. There was no significant difference between the two tested species of stick insects (*M. extradentata* and *S. inexpectata*) with intact claws despite different size of the claws.(III)Removing the claws showed better performance of attachment pads with nubby microstructures in face of wax-crystal-covered leaves that potentially contaminate the attachment pads.(IV)The long-lasting effects of the leaf surfaces on attachment were not evidenced. In summary, the attachment of Phasmatodea was affected by the different substrates, with rough surface and trichome coverage being beneficial for attachment. The wax crystals on the plant surface provided the highest potential for protection of the plant through decreasing the attachment of insects.

## Figures and Tables

**Figure 1 insects-13-00952-f001:**
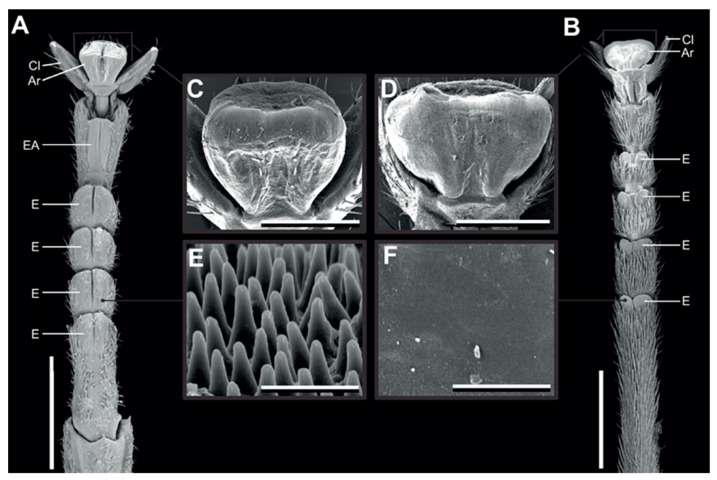
Tarsal morphology. Scanning electron micrographs of the feet of *Sungaya inexpectata* (**A**,**C**,**E**) and *Medauroidea extradentata* (**B**,**D**,**F**). (**A**,**B**) Ventral overviews. (**C**,**D**) Arolia. (**E**,**F**) microstructure of the euplantulae. Ar, arolium; EA, accessory euplantula (5th euplantula); E, euplantula; Cl, claw. Scale bars: 1 mm (**A**,**B**), 300 µm (**C**,**D**), 3 µm (**E**,**F**). Figure from Büscher and Gorb 2019 [26] reproduced with permission.

**Figure 2 insects-13-00952-f002:**
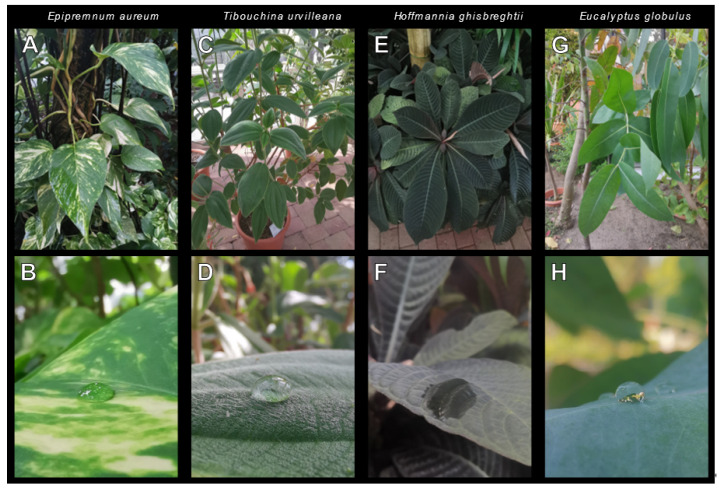
Plant habitus and a closeup of adaxial leaf surface with a waterdrop**.** (**A**,**B**) *Epipremnum aureum.* (**C**,**D**) *Tibouchina urvilleana* (**E**,**F**) *Hoffmannia ghiesbreghtii*. (**G**,**H**) *Eucalyptus globulus*. The contact angle of the droplet indicates the hydrophilicity/hydrophobicity of the leaf surface.

**Figure 3 insects-13-00952-f003:**
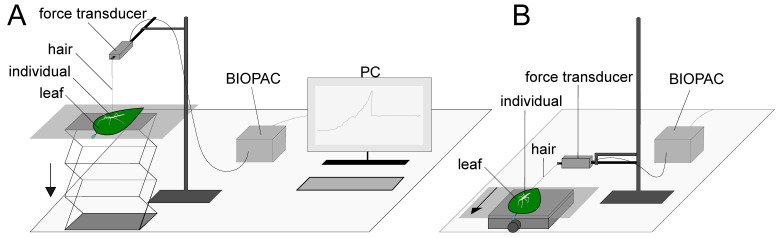
Experimental setup for the measurement of the pull-off force (**A**) and traction force (**B**). For simplification, the PC design is omitted in (**B**). The arrows indicate the direction in which the testing substrates were moved.

**Figure 4 insects-13-00952-f004:**
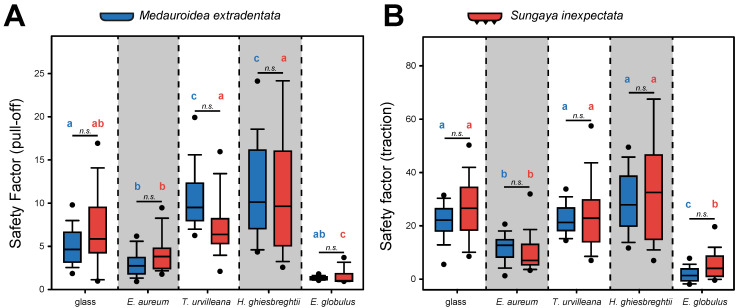
Force measurements on *M. extradentata* (blue) and *S. inexpectata* (red). (**A**,**B**) Safety factors: (**A**) pull-off force, (**B**) traction force. Statistically equal groups within each species are marked with the same letter and between the species with *n.s.* in comparison on the same substrate. Kruskal–Wallis one-way ANOVA on ranks, (*p* ≤ 0.001; pull-off: H = 287.832, d.f. = 9, *N* = 15; traction: H = 269.977, d.f. = 9, *N* = 45) and Tukey’s test (*p* < 0.05). The boxes indicate the 25th and 75th percentiles, whiskers define the 10th and 90th percentiles and the line within the boxes represents the median. Outliers are shown as individual points.

**Figure 5 insects-13-00952-f005:**
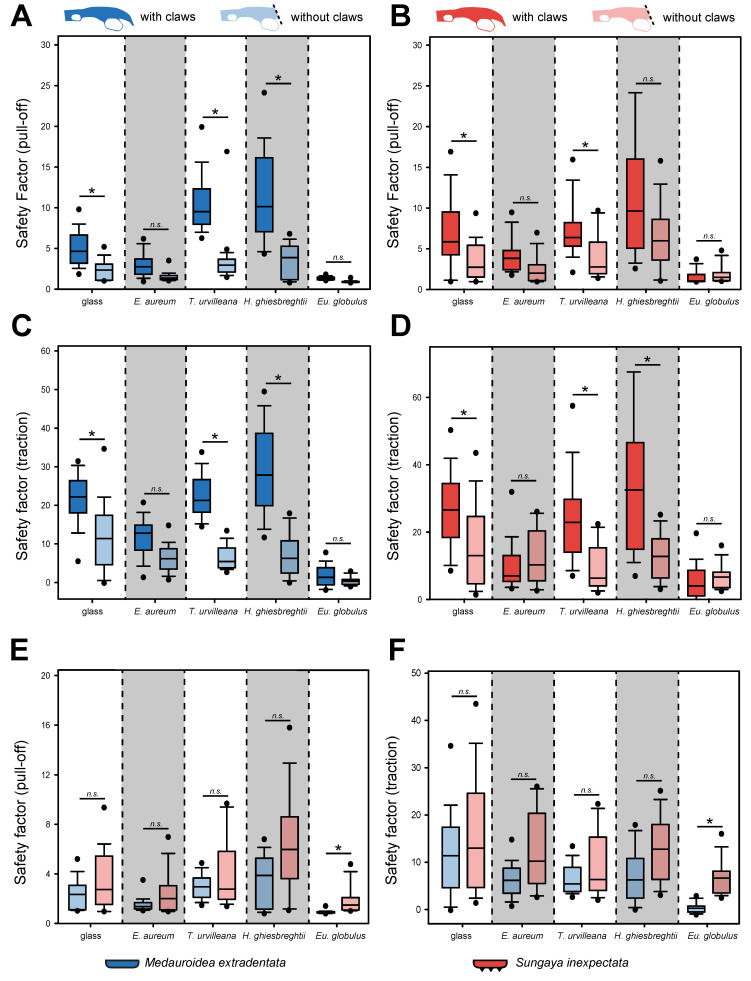
Safety factors of *M. extradentata* and *S. inexpectata* with claws and without claws on different substrates. (**A**) Pull-off force measurement of *M. extradentata* with and without claws. (**B**) Pull-off force measurement of *S. inexpectata* with and without claws. (**C**) Traction force measurement of *M. extradentata* with and without claws. (**D**) Traction force measurement of *S. inexpectata* with and without claws. (**E**) Pull-off force measurement of both species without claws. (**F**) Traction force measurement of both species without claws. Groups which do not differ significantly from each other are marked *n.s.* and groups which differ with a significance of *p* > 0.05 are marked * for each substrate. Kruskal–Wallis one-way ANOVA on ranks ((**A**): H = 287.832, (**B**): H = 276.410, (**C**): H = 179.328, (**D**): H =171.182, (**E**): H = 126.899, (**F**): H = 100.812; *p* ≤ 0.001, d.f. = 9, N_without_claws_ = 10, N_with_claws_ = 15) and Dunn’s method (**A**–**D**; *p* < 0.05) or Turkey’s test (**E**,**F**; *p* < 0.05). The boxes indicate 25th and 75th percentiles, whiskers define the 10th and 90th percentiles and the line within the boxes represents the median. Outliers are shown as individual points.

**Figure 6 insects-13-00952-f006:**
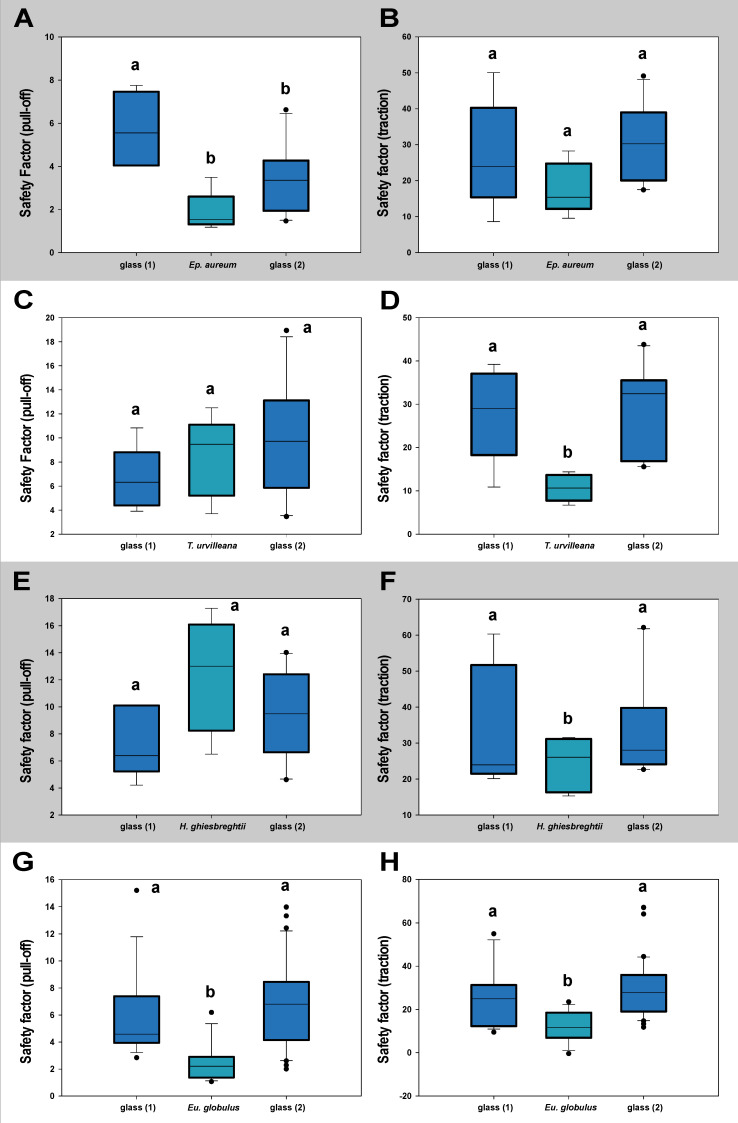
Contamination effect of the leaf substrates tested with *M. extradentata* shown through safety factor. (**A**) *Ep. aureum*, pull-off force. (**B**) *Ep. aureum,* traction force. (**C**) *T. urvilleana,* pull-off force. (**D**) *T. urvilleana* traction force. (**E**) *H. ghiesbreghtii,* pull-off force. (**F**) *H. ghiesbreghtii,* traction force. (**G**) *Eu. globulus,* pull-off force. (**H**) *Eu. globulus,* traction force. Statistically similar groups within a graph are marked with the same lowercase letter, tested through (**A**): one-way ANOVA (F = 8.453, d.f. = 2, N_glass1,Ep.aureum_ = 5, N_glass2_ = 10, *p* = 0.003) Holm–Šídák method (*p* < 0.05); (**B**): one-way ANOVA (F = 2.215, d.f. = 2, N_glass1,Ep.aureum_ = 5, N_glass2_ = 10, *p* = 0.140); (**C**): one-way ANOVA (F = 1.062, d.f. = 2, N_glass1,T.urvilleana_ = 5, N_glass2_ = 10, *p* = 0.368); (**D**): one-way ANOVA (F = 7.722, d.f. = 2, N_glass1,T.urvillena_ = 5, N_glass2_ = 10, *p* = 0.004), Holm–Šídák method (*p* < 0.05); (**E**): one-way ANOVA (F = 2.761, d.f. = 2, N_glass1,H.ghiesbreghtii_ = 5, N_glass2_ = 10, *p* = 0.092); (**F**): Kruskal–Wallis one-way ANOVA on ranks (H = 1.054, d.f. = 2, N_glass1,H.ghiesbreghtii_ = 5, N_glass2_ = 10, *p* = 0.590); (**G**): Kruskal–Wallis one-way ANOVA on ranks (H = 21.674, d.f. = 2, N_glass1,Eu.globulus_ = 15, N_glass2_ = 30, *p* ≤ 0.001), Dunn’s method (*p* < 0.05); (**H**): Kruskal–Wallis one-way ANOVA on ranks (H = 21.001, d.f. = 2, N_glass1,Eu.globulus_ = 15, N_glass2_ = 30, *p* ≤ 0.001), Dunn’s method (*p* < 0.05). The boxes indicate 25th and 75th percentiles, whiskers define the 10th and 90th percentiles and the line within the boxes represents the median. Outliers are shown as individual points.

**Figure 7 insects-13-00952-f007:**
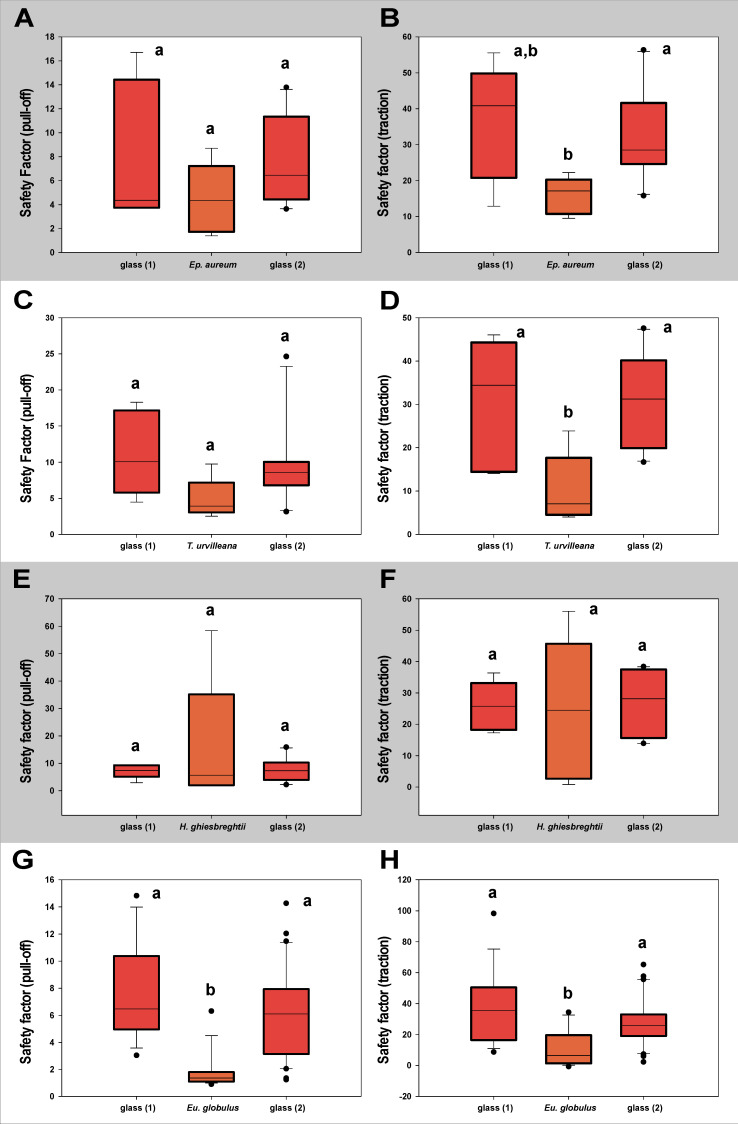
Contamination effect of the substrates tested with *S. inexpectata* shown through safety factor. (**A**) *Ep. aureum,* pull-off force. (**B**) *Ep. aureum,* traction force. (**C**) *T. urvilleana,* pull-off force. (**D**) *T. urvilleana,* traction force. (**E**) *H. ghiesbreghtii,* pull-off force. (**F**) *H. ghiesbreghtii,* traction force. (**G**) *Eu. globulus,* pull-off force. (**H**) *Eu. globulus,* traction force. Statistically similar groups within a graph are marked with the same lowercase letter. (**A**): Kruskal–Wallis one-way ANOVA on ranks (H = 1.054, d.f. = 2, N_glass1,Ep.aureum_ = 5, N_glass2_ = 10, *p* = 0.331); (**B**): one-way ANOVA (F = 4.013, d.f. = 2, N_glass1,Ep.aureum_ = 5, N_glass2_ = 10, *p* = 0.037; Holm–Šídák method, *p* < 0.05); (**C**): Kruskal–Wallis one-way ANOVA on ranks (H = 4.817, d.f. = 2, N_glass1,T.urvillena_ = 5, N_glass2_ = 10, *p* = 0.09); (**D**): one-way ANOVA (F = 5.927, d.f. = 2, N_glass1,T.urvillena_ = 5, N_glass2_ = 10, *p* = 0.011); Holm–Šídák method (*p* < 0.05); (**E**): one-way ANOVA (F = 5.927, d.f. = 2, N_glass1,T.urvillena_ = 5, N_glass2_ = 10, *p* = 0.011); Holm–Šídák method (*p* < 0.05); (**F**): Kruskal–Wallis one-way ANOVA on ranks (H = 0.189, d.f. = 2, N_glass1,H.ghiesbreghtii_ = 5, N_glass2_ = 10, *p* = 0.910); (**G**): Kruskal–Wallis one-way ANOVA on ranks (H = 25.805, d.f. = 2, N_glass1,Eu.globulus_ = 15, N_glass2_ = 30, *p* ≤ 0.001), Dunn’s method (*p* < 0.05); (**H**): Kruskal–Wallis one-way ANOVA on ranks (H = 17.517, d.f. = 2, N_glass1,Eu.globulus_ = 15, N_glass2_ = 30, *p* ≤ 0.001; Dunn’s method (*p* < 0.05). The boxes indicate 25th and 75th percentiles, whiskers define the 10th and 90th percentiles and the line within the boxes represents the median. Outliers are shown as individual points.

**Figure 8 insects-13-00952-f008:**
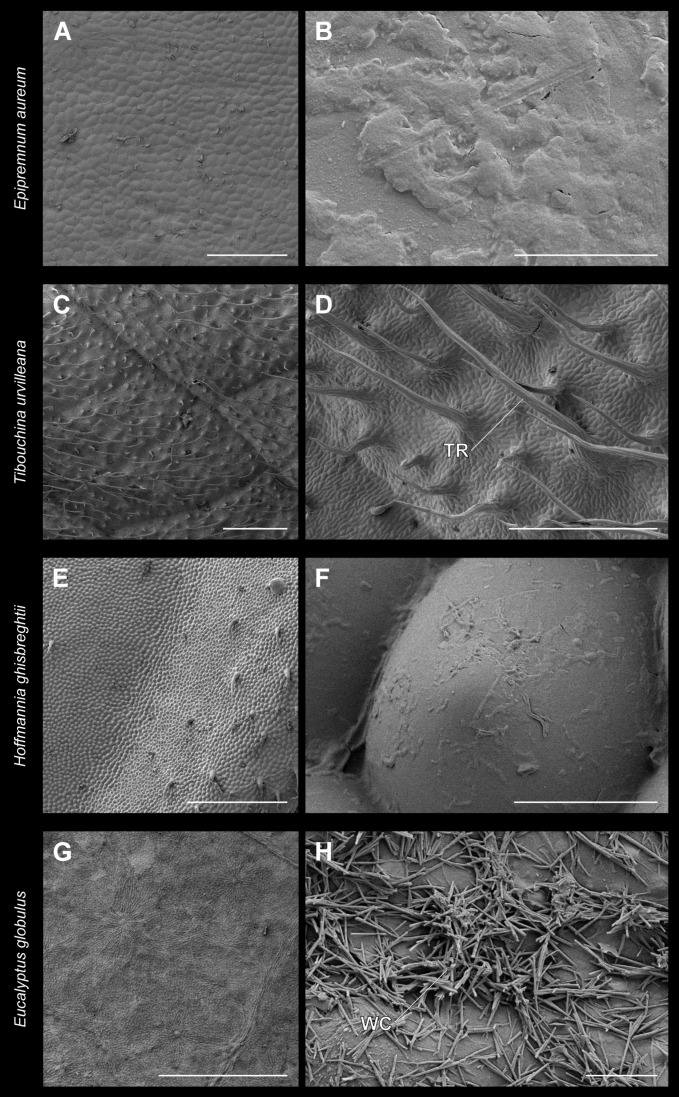
Cryoscanning electron microscopy images of the adaxial leaf surface. (**A**,**B**) *Epipremnum aureum*, (**C**,**D**) *Tibouchina urvilleana*, (**E**,**F**) *Hoffmannia ghiesbreghtii*, *(***G**,**H**) *Eucalyptus globulus*. Scale bars: (**A**,**D**) = 400 µm; (**B**,**H**) = 5 µm; (**C**,**E**,**G**) = 1 mm, (**F**) = 20 µm. TR: trichome; WC: wax crystals.

**Figure 9 insects-13-00952-f009:**
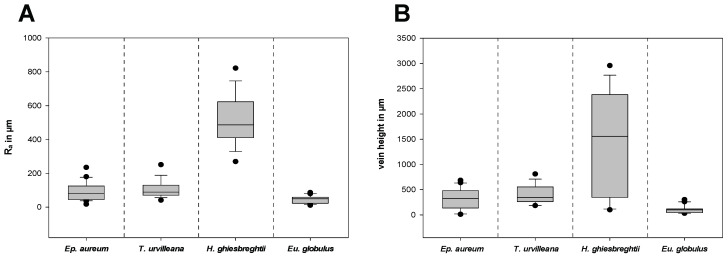
Characterization of the surface texture of the leaves. (**A**) Arithmetical mean roughness R_a_ and (**B**) vein height. *n* = 17 (*T. urvilleana*), 18 (*H. ghiesbreghtii*), 20 (*Ep. aureum*, *Eu. globulus*) for both (**A**,**B)**. The boxes indicate 25th and 75th percentiles, whiskers define the 10th and 90th percentiles and the line within the boxes represents the median. Outliers are shown as individual points.

## Data Availability

Raw data of the claw metrics, contamination measurements and leaf surface characterization are provided as Appendix A.

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
