# Peer review of "Attachment Performance of Stick Insects (Phasmatodea) on Plant Leaves with Different Surface Characteristics"

_insects, 2022, doi:10.3390/insects13100952_

Round 1
Reviewer 1 Report
General Comments: Be consistent with your terminology (don’t switch back-and-forth between terms, choose a term and stay with it). For example, sometimes you use “safety factor” other times you use pull off force or traction force. I suggest dropping “safety factor” because it is confusing. There is nothing in your manuscript related to safety.
Lines 133-134: milligrams might be better than grams for mass of your insects, but I don’t know what format the Journal prefers.
Line 172: ‘chosen’ instead of “picked out” and ‘weighed’ instead of “weighted”.
Lines 182-183: the curves were “visualized” and measured by the software, correct?
Lines 186-187: the “safety factor” is confusing. The term “safety” has no relevance to the experiments. Why not report you normalized ‘pull-off force’ by weights (mass) of the specimens, and once you have reported that you can still call it ‘pull-off force’. Why do you need to normalize the force by weight anyway? I wouldn’t think an insect’s weight would have much influence on ability to adhere to surfaces? I suppose size might because of differences in tarsal size and thus surface area for adherence.
Lines 188-189: did you test the same 15 individuals for each surface or test 15 individuals per surface? If you used the same 15 individuals, was a ‘resting period’ provided between surface tests? I don’t know if a ‘resting period’ would be necessary, I’m just trying to get a better picture of what was done.
Line 191: “The measurement” ‘The above measurements’?
Line 198: ‘each species’ instead of “the species”.
Lines 202-203: ‘previously described’ instead of “in the other force measurement experiments”.
Line 206: ‘were pooled” instead of “were pooled together” it’s redundant.
Line 213: delete “in five selected individuals of”, you already told us that in the previous line.
Line 220: delete “characteristics”.
RESULTS
General comments on Results. The Results text should be more concise. You don’t have to report the various force measurements in the text because you have them graphically illustrated in the Figures. Just report (in text) the treatments that were different not also those that were not different. For example, pull-off measurement was greater on H. ghiesbreghtii than glass, Ep. aureum, and Eu. globulus for M. extradentata, with no other differences among treatments (or some similar format). That type of ‘format’ will reduce the text considerably. Also, you can probably drop Kruskal–Wallis one-way ANOVA on Ranks from (Kruskal–Wallis one-way ANOVA on Ranks, H = 287.832, d.f. = 9, N = 45, P≤0.001; Tukey’s test, P<0.05). You already told us you were using Kruskal–Wallis one-way ANOVA on Ranks in the Methods, and I just realized you include it in the Figure captions. You only need it in one location unless this is the Journal’s format.
Fig. 4: Your lettering system is confusing. In 4A, you have a, b, c for the M. extradentata (blue) boxes. As a standardization, b should be intermediate between a and c, but it appears to represent the lowest ‘safety factor’. For S. inexpectata, you have boxes with ac designations but no c; since you have a, b, c, and d, b and c should be intermediate between a and d. N does not equal 45. You used 15 individuals and measured each one 3 times (lines 185, 188-189). The 3 measurements on the same individual are just sub-samples from which you calculate the mean or I suppose you could use the median. Either way that is just 15. Unless you used a different group of 15 individuals on each leaf surface (5 surfaces * 15), then n = 75 for each insect species.
Line 262: “12.190±4.840 (median±s.d.)”, how can you have a standard deviation for the median?
Fig. 5: Line 307—should be F) not D). You should state what the * represents and the level of significance (0.05 or 0.01 or ?).
Line 341: The term “without claws” is probably better than “ablated claws”. Whichever term you use be consistent don’t switch back-and-forth.
Line 375: Fig. 6 not 7.
Line 415: Fig. 7 not 8.
DISCUSSION
Check Discussion for proper tense, and try to reduce the amount of Introduction and Results that are repeated.
Author Response
Thank you very much for the feedback.
Referee: 1
General Comments: Be consistent with your terminology (don’t switch back-and-forth between terms, choose a term and stay with it). For example, sometimes you use “safety factor” other times you use pull off force or traction force. I suggest dropping “safety factor” because it is confusing. There is nothing in your manuscript related to safety.
Answer: We unified the terminology in the manuscript and now exclusively refer to pull-off and traction force to avoid confusion. Safety factor, however, is a well-established term, which expresses how many times an animal can attach its own weight to a surface, in attachment focused biomechanic studies and is used in hundreds of papers.
Lines 133-134: milligrams might be better than grams for mass of your insects, but I don’t know what format the Journal prefers.
Answer: Corrected. Grams changed to milligrams.
Line 172: ‘chosen’ instead of “picked out” and ‘weighed’ instead of “weighted”. Answer: Corrected.
Lines 182-183: the curves were “visualized” and measured by the software, correct?
Answer: Corrected.
Lines 186-187: the “safety factor” is confusing. The term “safety” has no relevance to the experiments. Why not report you normalized ‘pull-off force’ by weights (mass) of the specimens, and once you have reported that you can still call it ‘pull-off force’. Why do you need to normalize the force by weight anyway? I wouldn’t think an insect’s weight would have much influence on ability to adhere to surfaces? I suppose size might because of differences in tarsal size and thus surface area for adherence.
Answer: We now follow your suggestion and report that we normalized by mass and later on just call it accordingly (pull-off and traction force). Normalization in similar studies is often done by mass to correct for size differences. Both size and weight can potentially be used for normalization, but very often weight is chosen (see above regarding safety factors), as weight grows with a power of 3 (volume) in contrast to 2 for area. Pad area, however, is less informative in this example here, as the attachment microstructures of phasmids result in different real contact area for different species. The nubby AMS causes much less actual contact compared to the smooth one. Therefore the conventional normalization with the safety factor was chosen here. Also, it can be shown that in this system weight normalization is a better proxy than tarsal pad area correction (unpublished data).
Lines 188-189: did you test the same 15 individuals for each surface or test 15 individuals per surface? If you used the same 15 individuals, was a ‘resting period’ provided between surface tests? I don’t know if a ‘resting period’ would be necessary, I’m just trying to get a better picture of what was done.
Answer: The measurements were not always done with the same 15 individuals. Due to loss of legs while moulting or other cases of dysfunctionality some Phasmids were replaced between measurements. A sufficient resting time between the measurements was ensured. This is now noted in the Material and Method section.
Line 191: “The measurement” ‘The above measurements’?
Answer: Corrected.
Line 198: ‘each species’ instead of “the species”.
Answer: Corrected.
Lines 202-203: ‘previously described’ instead of “in the other force measurement experiments”.
Answer: Corrected.
Line 206: ‘were pooled” instead of “were pooled together” it’s redundant. Answer: Corrected.
Line 213: delete “in five selected individuals of”, you already told us that in the previous line. Answer: Corrected. Line 220: delete “characteristics”.
Answer: Corrected.
RESULTS General comments on Results.
The Results text should be more concise. You don’t have to report the various force measurements in the text because you have them graphically illustrated in the Figures. Just report (in text) the treatments that were different not also those that were not different. For example, pull-off measurement was greater on H. ghiesbreghtii than glass, Ep. aureum, and Eu. globulus for M. extradentata, with no other differences among treatments (or some similar format). That type of ‘format’ will reduce the text considerably. Also, you can probably drop Kruskal–Wallis one-way ANOVA on Ranks from (Kruskal–Wallis one-way ANOVA on Ranks, H = 287.832, d.f. = 9, N = 45, P≤0.001; Tukey’s test, P<0.05). You already told us you were using Kruskal–Wallis one-way ANOVA on Ranks in the Methods, and I just realized you include it in the Figure captions. You only need it in one location unless this is the Journal’s format.
Answer: We have shortened the result section where applicable. We omitted the repetition of the statistic test where possible. In some paragraphs we still kept it to be precise with the report of the specific tests, as one-way ANOVAS were used in some comparisons due to the normality of the data.
Fig. 4: Your lettering system is confusing. In 4A, you have a, b, c for the M. extradentata (blue) boxes. As a standardization, b should be intermediate between a and c, but it appears to represent the lowest ‘safety factor’. For S. inexpectata, you have boxes with ac designations but no c; since you have a, b, c, and d, b and c should be intermediate between a and d. N does not equal 45. You used 15 individuals and measured each one 3 times (lines 185, 188-189). The 3 measurements on the same individual are just sub-samples from which you calculate the mean or I suppose you could use the median. Either way that is just 15. Unless you used a different group of 15 individuals on each leaf surface (5 surfaces * 15), then n = 75 for each insect species.
Answer: The lettering system follows the reading direction from left to right. We believe that this sequence is as intuitive as ranked by height of the boxes. We briefly checked other studies and found both ways for lettering, but more frequently the sequence from left to right. “ac” in 4A is indeed not correct, the “c” is not needed. N is now corrected as well.
Line 262: “12.190±4.840 (median±s.d.)”, how can you have a standard deviation for the median?
Answer: Corrected.
We now report mean ±s.d. Fig. 5: Line 307—should be F) not D). You should state what the * represents and the level of significance (0.05 or 0.01 or ?). Answer: Corrected. * represents P<0.05.
Line 341: The term “without claws” is probably better than “ablated claws”. Whichever term you use be consistent don’t switch back-and-forth.
Answer: Corrected. The term “without claws” is now used.
Line 375: Fig. 6 not 7.
Answer: Corrected.
Line 415: Fig. 7 not 8.
Answer: Corrected.
DISCUSSION
Check Discussion for proper tense, and try to reduce the amount of Introduction and Results that are repeated.
Answer: We checked the discussion for proper tense and shortened it by omitting repetitions.
Reviewer 2 Report
It was with pleasure that I read the present study on the interaction of the stick insect attachment system and natural leaf surfaces. After years of extensive studies of biological attachment systems on artificial substrates we have gained enough knowledge of the physical principles behind these attachment systems that we can start to look at the interaction of these systems with natural surfaces. Such studies are of prime importance to understand the evolutionary and ecological drivers of natural attachment systems. In so far, this study is well timed.
I only have a few points I would like to raise to the attention of the authors, which hopefully will further improve the MS.
The introduction is generally well written, and I mostly have only a few minor comments to improve the clarity and the language here. One more general point concerns the different parts of the adhesive system. You mention all three parts, arolium, euplantulae and paired claws in the beginning, but then you focus only on the euplantulae and how differences here affect adhesion. One though for consideration is that the implicit reason for this focus is that this part shows structural variation among species, while the others apparently do not do so, right? So, maybe you could add one sentence to make this as an explicit statement to make it easier for the reader to follow your logic. Another, and in my opinion more important aspect is that since you experimentally test the influence of claw removal on the attachment abilities, you should expand the introduction by explaining what is known about the importance and contribution of the claws to the adhesive performance in stick insects.
The methods are appropriate for the research question. But I do have two general comments/questions:
1) For the force measurements you use a nested design with 15 specimens measured three times each. But in the statistical analysis section you do not state explicitly, if/how you accounted for repeated measurements. It may seem obvious to the authors that this has been done, but for the reader to assess the validity of your approach it would be good to add this information.
2) The experiment on the potential contamination effects appears relatively preliminary to me. This is totally fine, but I am wondering why the authors expect to find potential contamination on a scale that affect the animals already after a single pull on a plant substrate. I personally would expect contamination to accumulate over time and am not surprised that you don’t find much contamination effects after a single pull. The MS is already quite comprehensive, so I would not expect an additional full-scale experiment like testing animals repeatedly on glass and on different substrates to measure differences in decline on the different substrates. But still I feel that some rational could be provided on what grounds the authors did or did not expect contamination from a single measurement. If, on the other hand, this is purely exploratory without any prior expectations, then the authors should highlight this more in the introduction and temper their language when interpreting the respective results in the discussion.
In the results section you repeatedly report significant differences among categories without writing which of the two groups performed better or worse. This information is partially only available from the figures. Please rewrite these sentences accordingly thought the section.
Finally, in the introduction the authors mention other studies dealing with the effects of natural plant surfaces on the attachment of other insect species, but they do not come back to these studies in the discussion. I think that the paper would highly benefit, if the authors could add a paragraph, where they discuss their own findings for stick insects in comparison with the results of these studies on other clades. This could eventually even be extended to studies on non-insect clades, although this last thought might go beyond the scope of the journal and/or the MS.
Detailed comments:
Simple Summary and Abstract:
L13: As I am coming from the vertebrate world, I am not sure how common these terms are, but I think that at least for a simple summary the terms ‘arolium’ and ‘euplantulae’ should either be defined or described when they first appear.
L31: You mention both, the arolium and the euplantulae as part of the attachment system. But then you focus completely on the microstructures of the euplantulae without mentioning the arolium anymore. From reading only the abstract it is unclear why this is the case. Please elaborate shortly on the reason for focussing on the euplantulae in the abstract.
L34: …representing different types among THE high diversity of plant surfaces.
L37: Was ‘loss of mechanical interlocking’ really observed? The way I understood the paper, you observed lower attachment due to claw removal and then conclude that this is due to loss of mechanical interlocking. Please reformulate accordingly.
Introduction:
L46: ‘interacted for’ or ‘interact since’
L83: …species and A worldwide distribution.
L91-93: could you elaborate more on how this complementary mechanism is generated and uphold? Is this due to leg morphology per se or does it result from behavior control of the legs?
L95: Are there any studies clearly showing that this variation in surface microstructures is an adaptation to different substrate conditions? I would agree that this is most likely the case, but unless there are studies showing this (in which case they should be cited here) I would suggest tampering the language here by adding ‘most likely’ (or something similar) before ‘to adapt best…’.
L99-104: I do not see a reason why Büscher and Grob [26] needs to be highlighted in this sentence. In my opinion, the sentence could be condensed e.g., “The adhesion and traction of the nubby and smooth euplantulae have been already experimentally tested on artificial surfaces with different roughness. These studies showed that smooth euplantular microstructures provide better attachement […] [26,27,31].” If there is something that makes the study 26 standing out, this should be made explicit.
L117: better use ‘explored’ then ‘tested’ here (see my general comment on the contamination experiment).
Methods:
L120: This should be Species not Specimens.
L124 (Fig.1): What is the accessory euplantula? The term only occurs here and is not defined, explained or mentioned anywhere else in the MS. Either remove it from the figure or explain/define it.
L148: please provide a short definition of the term ‘lamina’ for readers unfamiliar with morphology.
L212/213 ‘…fife selected individuals…’: selected how? It you had criteria after which you selected the individuals (maybe you checked them for intact claws?) please name them. Otherwise, the term selected is meaningless here and can be omitted.
L225: The supplementary material should be ordered following its referencing in the MS. Unless I overlooked something this is the first reference to supplementary material, so this should be SI1. Also, all supplementary materials need to be referenced in the MS, which is not the case yet.
L232 ‘See Gorb and Gorb (2009)’: this is an inconsistent reference. It should be ‘Gorb and Gorb [#]’.
Results:
L258-260: Are these differences significant?
L280-282: “On E. globulus, the lowest traction was obtained with a median safety factor of…” NOT “On E. globulus, the lowest traction with a median safety factor of […] was obtained”
L284-287: This sentence is a bit difficult to follow. I would suggest changing it to something along the lines of ‘For M. extradentata there were no significant differences among glass, T. urvilleana and H. ghisbreghtii.’
L317 “The highest median safety factor was similar in the measurements with claws…”: similar to what? This is unclear to me.
L326 'median of the safety factors ': what does this mean? The median of both measurements combined? How is that supporting the absence of differences?
L421 ‘were statistically not different’: read ‘were not statistically different’
L430-432: This additional control measurement needs to be described in the methods.
L435 - Fig.8: The figure could be improved by placing the units of the scale bar in the images above the scale bars.
Discussion:
L527: what are statistically visible differences? Please change to (statistically) significant.
L577: independent NOT independently
L583: less influential than what? I assume you mean less than smooth structures, but then please say so.
L610: ‘could BE a result of’
Author Response
It was with pleasure that I read the present study on the interaction of the stick insect attachment system and natural leaf surfaces. After years of extensive studies of biological attachment systems on artificial substrates we have gained enough knowledge of the physical principles behind these attachment systems that we can start to look at the interaction of these systems with natural surfaces. Such studies are of prime importance to understand the evolutionary and ecological drivers of natural attachment systems. In so far, this study is well timed.
I only have a few points I would like to raise to the attention of the authors, which hopefully will further improve the MS.
Answer: Thank you for the constructive feedback and thoroghfull review.
The introduction is generally well written, and I mostly have only a few minor comments to improve the clarity and the language here. One more general point concerns the different parts of the adhesive system. You mention all three parts, arolium, euplantulae and paired claws in the beginning, but then you focus only on the euplantulae and how differences here affect adhesion. One though for consideration is that the implicit reason for this focus is that this part shows structural variation among species, while the others apparently do not do so, right? So, maybe you could add one sentence to make this as an explicit statement to make it easier for the reader to follow your logic. Another, and in my opinion more important aspect is that since you experimentally test the influence of claw removal on the attachment abilities, you should expand the introduction by explaining what is known about the importance and contribution of the claws to the adhesive performance in stick insects.
Answer: A sentence about the focus on the euplantula is now added and the importance of claws highlighted in the introduction.
The methods are appropriate for the research question. But I do have two general comments/questions:
1) For the force measurements you use a nested design with 15 specimens measured three times each. But in the statistical analysis section you do not state explicitly, if/how you accounted for repeated measurements. It may seem obvious to the authors that this has been done, but for the reader to assess the validity of your approach it would be good to add this information.
Answer: The repeated measurements of single individuals on the same substrate were averaged to reduce intraindividual variability. The means obtained from these measurement were then compared between the treatments. During the measurements we randomized the sequence of the individuals per substrate and the different substrates were used on different days. We now clarified these explanations in the manuscript.
2) The experiment on the potential contamination effects appears relatively preliminary to me. This is totally fine, but I am wondering why the authors expect to find potential contamination on a scale that affect the animals already after a single pull on a plant substrate. I personally would expect contamination to accumulate over time and am not surprised that you don’t find much contamination effects after a single pull. The MS is already quite comprehensive, so I would not expect an additional full-scale experiment like testing animals repeatedly on glass and on different substrates to measure differences in decline on the different substrates. But still I feel that some rational could be provided on what grounds the authors did or did not expect contamination from a single measurement. If, on the other hand, this is purely exploratory without any prior expectations, then the authors should highlight this more in the introduction and temper their language when interpreting the respective results in the discussion.
Answer: As shown for other insects, wax crystals often contaminate the attachment system at first contact and are subsequently removed/cleaned off during the next footsteps. As we expected such an effect for one plant this was done for the species of concern, but added for all other species in an exploratory manner. This information is now clarified in the manuscript.
In the results section you repeatedly report significant differences among categories without writing which of the two groups performed better or worse. This information is partially only available from the figures. Please rewrite these sentences accordingly thought the section.
Answer: We added this information.
Finally, in the introduction the authors mention other studies dealing with the effects of natural plant surfaces on the attachment of other insect species, but they do not come back to these studies in the discussion. I think that the paper would highly benefit, if the authors could add a paragraph, where they discuss their own findings for stick insects in comparison with the results of these studies on other clades. This could eventually even be extended to studies on non-insect clades, although this last thought might go beyond the scope of the journal and/or the MS.
Answer: A different reviewer suggested to reduce the repetitions of content included in the introduction already. We now tried to find a compromise between the two suggestions.
Detailed comments:
Simple Summary and Abstract:
L13: As I am coming from the vertebrate world, I am not sure how common these terms are, but I think that at least for a simple summary the terms ‘arolium’ and ‘euplantulae’ should either be defined or described when they first appear.
Answer: Corrected.
L31: You mention both, the arolium and the euplantulae as part of the attachment system. But then you focus completely on the microstructures of the euplantulae without mentioning the arolium anymore. From reading only the abstract it is unclear why this is the case. Please elaborate shortly on the reason for focussing on the euplantulae in the abstract.
Answer: Corrected.
L34: …representing different types among THE high diversity of plant surfaces.
Answer: Corrected.
L37: Was ‘loss of mechanical interlocking’ really observed? The way I understood the paper, you observed lower attachment due to claw removal and then conclude that this is due to loss of mechanical interlocking. Please reformulate accordingly.
Answer: Corrected.
Introduction:
L46: ‘interacted for’ or ‘interact since’
Answer: Corrected
L83: …species and A worldwide distribution.
Answer: Corrected.
L91-93: could you elaborate more on how this complementary mechanism is generated and uphold? Is this due to leg morphology per se or does it result from behavior control of the legs?
Answer: Corrected. Attachment functions of the pads now described in more detail.
L95: Are there any studies clearly showing that this variation in surface microstructures is an adaptation to different substrate conditions? I would agree that this is most likely the case, but unless there are studies showing this (in which case they should be cited here) I would suggest tampering the language here by adding ‘most likely’ (or something similar) before ‘to adapt best…’.
Answer: Corrected.
L99-104: I do not see a reason why Büscher and Grob [26] needs to be highlighted in this sentence. In my opinion, the sentence could be condensed e.g., “The adhesion and traction of the nubby and smooth euplantulae have been already experimentally tested on artificial surfaces with different roughness. These studies showed that smooth euplantular microstructures provide better attachement […] [26,27,31].” If there is something that makes the study 26 standing out, this should be made explicit.
Answer: Corrected.
L117: better use ‘explored’ then ‘tested’ here (see my general comment on the contamination experiment).
Answer: Corrected.
Methods:
L120: This should be Species not Specimens.
Answer: Corrected
L124 (Fig.1): What is the accessory euplantula? The term only occurs here and is not defined, explained or mentioned anywhere else in the MS. Either remove it from the figure or explain/define it.
Answer: We added the information to the manuscript.
L148: please provide a short definition of the term ‘lamina’ for readers unfamiliar with morphology.
Answer: Corrected.
L212/213 ‘…fife selected individuals…’: selected how? It you had criteria after which you selected the individuals (maybe you checked them for intact claws?) please name them. Otherwise, the term selected is meaningless here and can be omitted.
Answer: Corrected.
L225: The supplementary material should be ordered following its referencing in the MS. Unless I overlooked something this is the first reference to supplementary material, so this should be SI1. Also, all supplementary materials need to be referenced in the MS, which is not the case yet.
Answer: Corrected.
L232 ‘See Gorb and Gorb (2009)’: this is an inconsistent reference. It should be ‘Gorb and Gorb [#]’.
Answer: Corrected.
Results:
L258-260: Are these differences significant?
Answer: We did not test the significance of the difference between pull-off and traction force, it was just a general notion that the traction is always higher than the pull-off force.
L280-282: “On E. globulus, the lowest traction was obtained with a median safety factor of…” NOT “On E. globulus, the lowest traction with a median safety factor of […] was obtained”
Answer: Corrected.
L284-287: This sentence is a bit difficult to follow. I would suggest changing it to something along the lines of ‘For M. extradentata there were no significant differences among glass, T. urvilleana and H. ghisbreghtii.’
Answer: Corrected
L317 “The highest median safety factor was similar in the measurements with claws…”: similar to what? This is unclear to me.
Answer: We slightly rephrased the sentence to make it clear.
L326 'median of the safety factors ': what does this mean? The median of both measurements combined? How is that supporting the absence of differences?
Answer: Corrected.
L421 ‘were statistically not different’: read ‘were not statistically different’
Answer: Corrected.
L430-432: This additional control measurement needs to be described in the methods.
Answer: Corrected.
L435 - Fig.8: The figure could be improved by placing the units of the scale bar in the images above the scale bars.
Answer: We prefer to have the scale in the figure caption. The units of each scale bar are described in the caption of the figure to minimize the distraction within the images.
Discussion:
L527: what are statistically visible differences? Please change to (statistically) significant.
Answer: Corrected.
L577: independent NOT independently
Answer: Corrected
L583: less influential than what? I assume you mean less than smooth structures, but then please say so.
Answer: Corrected
L610: ‘could BE a result of’
Answer: Corrected
Reviewer 3 Report
The manuscript is well structured with logic and sound results. However, concerns need to be addressed before the publication:
1. Author might need to remove the "simple summary" section, which is redundant to the "Abstract".
2. Previously, it has been shown that ambient humidity could affect the attachment ability of insects (e.g., Beilstein J. Nanotechnol. 2016, 7, 1322–1329. ). The authors indicated that the experiments were performed under ambient humidity of 17-52%. The authors should evaluate the possible influence of ambient humidity on the attachment ability of stick insects. Or, it would be difficult to interpret the experimental results correctly under the variation of ambient humidity.
Author Response
Referee 2:
The manuscript is well structured with logic and sound results. However, concerns need to be addressed before the publication:
Author might need to remove the "simple summary" section, which is redundant to the "Abstract".
Answer: We agree, but the simple summary section is mandatory in this journal. We requested to omit this section, but the editorial offices insists to keep it.
Previously, it has been shown that ambient humidity could affect the attachment ability of insects (e.g., Beilstein J. Nanotechnol. 2016, 7, 1322–1329. ). The authors indicated that the experiments were performed under ambient humidity of 17-52%. The authors should evaluate the possible influence of ambient humidity on the attachment ability of stick insects. Or, it would be difficult to interpret the experimental results correctly under the variation of ambient humidity.
Answer: We found, similar to previous studies (https://doi.org/10.1242/jeb.209833), no indication for an effect of ambient humidity in this range for stick insects. We now refer to this in the manuscript.
Round 2
Reviewer 3 Report
The authors have modified the manuscript appropriately. I think the manuscript is suitable for publication in its present form.
Author Response
Thank you very much.